# Contactless Stethoscope Enabled by Radar Technology

**DOI:** 10.3390/bioengineering10020169

**Published:** 2023-01-28

**Authors:** Isabella Lenz, Yu Rong, Daniel Bliss

**Affiliations:** The Center for Wireless Information Systems and Computational Architectures (WISCA), Arizona State University, Tempe, AZ 85281, USA

**Keywords:** cardiograph, heart sound, millimeter wave, MIMO, radar, signal processing

## Abstract

Contactless vital sign measurement technologies have the potential to greatly improve patient experiences and practitioner safety while creating the opportunity for comfortable continuous monitoring. We introduce a contactless alternative for measuring human heart sounds. We leverage millimeter wave frequency-modulated continuous wave radar and multi-input multi-output beamforming techniques to capture fine skin vibrations that result from the cardiac movements that cause heart sounds. We discuss contact-based heart sound measurement techniques and directly compare the radar heart sound technique with these contact-based approaches. We present experimental cases to test the strengths and limitations of both the contact-based measurement techniques and the contactless radar measurement. We demonstrate that the radar measurement technique is a viable and potentially superior method for capturing human heart sounds in many practical settings.

## 1. Introduction

An estimated 17.9 million people die from cardiovascular disease every year, making it the leading cause of death worldwide. Cardiac timing intervals provide insight into the heart’s function, which aids cardiovascular disease diagnosis and risk assessment [1]. Most conventional clinical cardiac measurement methods are contact-based. This can be seen in Figure 1. This can be uncomfortable and inconvenient for both patients and practitioners, and is not practical for long-term continuous monitoring. The close patient–practitioner proximity needed to collect these measurements increases the risk of spreading communicable diseases [2]. Additionally, only one patient can be monitored at a time. Contactless measurement can address these challenges. Radar has been explored extensively as a contactless cardiac monitoring alternative.

Conceptually, radar systems capture cardiac activity by transmitting an electromagnetic waveform towards a subject and then measuring the reflected wave. The reflected wave is modulated by the movement of the chest wall induced by the heart’s movements [3]. Many different radar waveforms have demonstrated success in measuring heart rate, including continuous wave Doppler [4,5]; ultra-wideband impulse [6,7,8]; frequency-modulated, continuous-wave (FMCW) [9,10,11]; and stepped-frequency, continuous-wave (SFCW) [12,13]. In combination with these waveforms, a variety of signal processing approaches have been used to extract the heart rate from the received signal, including time–frequency analysis [14], numerical analysis [15], autoregression algorithms [2], ranging and localization approaches [16], motion cancellation [17], and algorithms based on mathematical and experimental modeling [18].

To date, most radar-based cardiac monitoring approaches have only been able to recover the fundamental heart beat. There are, however, more cardiac events of clinical interest than the heart beat. One such event is the heart sound. During one cardiac cycle, blood first collects in the upper left and right atria of the heart. The atria walls then contract and expel blood into the lower left and right ventricles. The valves that separate the atria and ventricles open and close to allow blood to pass through; this causes the first heart sound, S1. The walls of the ventricles then contract, and blood is expelled from the left ventricle to the rest of the body and from the right ventricle to the lungs. The pulmonary and aortic valves open and close, causing the second heart sound, S2, and the cycle starts over [19]. This cycle is depicted in Figure 2.

There are two conventional methods of monitoring heart sounds, both of which are contact-based. The most common method is to use a stethoscope. A stethoscope consists of a disc-shaped resonator and a tube that connects to two ear pieces. The resonator captures acoustic pressure waves generated by vibrations of the skin that occur as a result of the heart valves closing. The waves then travel through the tubes to the ear pieces. There are also electronic stethoscopes that convert the the acoustic waves to digital signals that can be amplified for optimal listening, or wirelessly collected for remote diagnosis and data storage [21]. The second common method of heart sound measurement is a seismocardiogram. This measurement is collected by attaching a small accelerometer to the patients chest. The accelerometer captures the vibrations of the skin caused by the heart valves closing [22]. Figure 3 depicts the contact-based and contactless radar measurement setups.

There is motivation for a contactless heart sound measurement method. Exploration of and success with radar for this purpose is, however, extremely limited. This is because the vibrations of the skin caused by the heart valves closing are minute, in the order of a few micrometers [23]. Radar technologies have improved in recent years, and mmWave radar systems are now available. The higher operating frequencies offer increased fractional bandwidth, which improves range resolution, and the shorter wavelengths increase phase sensitivity. This allows for skin vibrations to be captured using phase-based methods [24].

In this paper, we capitalize on radar technology improvements and use mmWave FMCW and advanced MIMO radar beamforming techniques to capture vibrations of the skin caused by heart sounds. We synchronize this measurement with the conventional contact-based stethoscope and accelerometer methods for direct comparison of signal quality and features. We then present a series of experimental cases in which we demonstrate the advantages and shortcomings of the radar measurement, stethoscope, and accelerometer methods.

This paper is organized as follows: In Section 2, we present the commercial off-the-shelf devices used to collect the radar and reference heart sound measurements. In Section 3, we discuss the signal processing steps used to extract the heart sound signatures from the raw data collected by the contact-based stethoscope and accelerometer methods and the contactless radar system. In Section 4, we layout the experimental setup for the reference and challenge cases used in our data collection. In Section 5, we explain how we analyzed the heart sound signal features and quality. In Section 6, we present the heart sound signals and our analysis of the signals for each of the test cases. Finally, in Section 7 and Section 8, we provide concise commentary on our most significant study results and challenges, and draw conclusions regarding our overall contributions.

Our Contributions:Novel radar heart sound measurement described in detail for reproducibility;Time-synchronized comparison of radar, stethoscope, and accelerometer heart sound measurements;Experimental cases to test viability and limitations of radar measurement compared to conventional stethoscope and accelerometer methods.

## 2. Measurement Systems

We use four systems to measure heart sound in our studies, our proposed novel radar system and three conventional contact-based measurement systems. Visual depictions of the systems can be seen in Figure 4. A breakdown of key features of each system can be found in Table 1.

### 2.1. Radar

We use a mmWave FMCW MIMO radar sensor, as seen in Figure 4a (Texas Instruments AWR6843 Radio Frequency (RF) Array and DCA1000EVM Data Capture Adapter [25]), to capture cardiac activity. The block diagram of the RF array and transmit signal can be seen in Figure 5. We configure the FMCW chirp signal to have a 77 GHz starting frequency ($f_{0}$) and 3.6 GHz bandwidth (B). The time of each chirp is 60 microseconds. We use the sensor’s three transmit (TX) channels. Each frame consists of three chirps transmitted in TDM fashion, meaning only one channel is transmitting at a given time, while the other channels are switched off. The frame interval ($T_{FI}$) is 1 millisecond. The four receive (RX) channels are used complete the MIMO phased array for beamforming. We mount the sensor on a common camera tripod.

We assess the potential health hazards of the system. According to the ICNIRP, above 6 GHz, it is useful to describe exposure in terms of density of absorbed power per area, because electromagnetic fields are absorbed superficially in this frequency range. The maximum transmit power of the radar system is 12 dBm. We assume no transmit power loss for the sake of simplicity. We approximate the subject’s torso area as 30 cm by 30 cm. The density of the absorbed power per area is therefore 0.167 W/m^2^. This level is significantly below the ICNIRP general public safe exposure level of 10 W/m^2^ in the 2–300 GHz frequency range [26].

### 2.2. Accelerometer

We use the accelerometer signal as the second conventional reference. The accelerometer captures cardiac vibrations by attaching a light-weight sensor directly on the chest surface. We use a three-axis accelerometer (TDK MPU-6050 [27]) in combination with a microcontroller (Arduino UNO R3 [28]), as seen in Figure 4b. The sensor’s sensitivity is ±2 g-forces. We capture and process the *Z*-axis acceleration data. We attach the accelerometer to the subject using an elastic band that wraps around the back of the chest.

### 2.3. Stethoscope

We use an acoustic stethoscope to collect the first reference heart sound measurement. We use a conventional analog stethoscope head. The headpiece shown in Figure 4c contains a resonator that captures the acoustic pressure waves. The raw analog heart sound signal without any pre-filtering or amplification is recorded by connecting to an audio interface (Behringer U-Phoria UMC22 USB Audio Interface) for data acquisition.

Modern digital stethoscopes are equipped with amplification and filtering devices. This improves sound quality, addresses humans’ limited ability to hear low frequencies, and provides a visual representation of the signal [29,30,31]. In this study, we also include data from a state-of-the-art digital stethoscope from [32], as shown in Figure 4d, for one of our test cases. The goal is to not only use the digital stethoscope as a heart sound reference, but also test its performance, compared to the acoustic stethoscope, in real-world cases. We find that the acoustic and digital stethoscope generate very similar measurement results in all the test cases except the included case.

## 3. Signal Processing

We present the signal processing steps used for each measurement approach. A summary of the steps can be found in Figure 6.

### 3.1. Stethoscope Signal Processing

The digital converter applies a built-in 20 Hz high-pass filter. We resample the signal to 1000 Hz to match the radar sampling frequency.

### 3.2. Accelerometer Signal Processing

We use a two-step processing approach to extract the heart sounds from the raw accelerometer signal. In the first step, we apply a high-pass filter with a 20 Hz corner frequency. This removes the fundamental heart beat and respiration frequencies from the signal. We then apply a moving average over 20 samples to denoise the signal. This process can be seen in Figure 7.

### 3.3. Radar Signal Processing

We use FMCW signal processing to select the range bin with strongest return. Further explanation of our approach can be found in [23]. In this paper, we measure heart sound on one subject; however, the MIMO configuration and beamforming approach are conducive to multi-subject measurement. The multiple received spatial channels are digitally steered in the direction of the subject’s chest to enhance measurement signal quality. The beam steering vector is given as
(1)$$w={\left[1\;\;e^{j\frac{2\pi d_{0}}{\lambda }\sin\left(\theta \right)}\;\;e^{j\frac{2\pi d_{0}}{\lambda }2\sin\left(\theta \right)}....e^{j\frac{2\pi d_{0}}{\lambda }\left(N-1\right)\sin\left(\theta \right)}\right]}^{T}$$
where θ denotes target spatial angle, λ is wavelength, and $d_{0}$ is array element spacing. The N-element array output is given as
(2)$$S=\left[s_{1}\;\;s_{2}...s_{N}\right]$$
where each channel, $s_{n}$, in the baseband takes the form
(3)$$s_{n}\left(\tau ,t\right)=Ae^{j\left[2\pi \alpha \tau_{d}\left(t\right)\tau +2\pi f_{c}\tau_{d}\left(t\right)\right]}\times e^{j\left[\frac{2\pi d_{0}}{\lambda }\left(n-1\right)\sin\left(\theta \right)\right]}$$
where *A* is a complex number that represents the round trip path loss, τ and *t* denote fast time and slow time, respectively, α is the frequency slope rate, and $f_{c}$ is the sweep starting frequency. The delay is time-varying, and a function of $D_{p}\left(t\right)$, the human chest motion. The delay takes the form $\tau_{d}\left(t\right)=2D_{p}\left(t\right)/c$. The beamformer output is given as
(4)$$\begin{array}{cc} y(\tau ,t)= & Sw^{*}=NAe^{j\left[2\pi \alpha \tau_{d}\left(t\right)\tau +2\pi f_{c}\tau_{d}\left(t\right)\right]}. \end{array}$$
Next, range processing is performed. The beat frequency, $f_{b}=\alpha \tau_{d}\left(t\right)$, is used to generate the range profile via fast Fourier transform (FFT) with respect to τ.
(5)$$y\left(\nu ,t\right)=NA\delta \left(\nu -f_{b}\right)\;\;e^{j4\pi \frac{D_{0}+D_{p}\left(t\right)}{\lambda }},$$
where ν denotes the FFT of *t* and $\lambda =c/f_{c}$ is the wavelength. The strongest peak in the range profile is selected as the target of interest, and (Equation 5) is reduced to
(6)$$y\left(f_{b},t\right)=NAe^{j\frac{4\pi D_{p}\left(t\right)}{\lambda }}$$
The phase of Equation (Equation 6) varies with time, and is linearly related to the skin vibration, $D_{p}\left(t\right)$. A high-pass filter, with a cut-off frequency of 20 Hz, is applied to the unwrapped phase signal to filter out chest movement from heart contraction and recover the heart sound signal. The results of each step in the radar signal processing procedure and a comparison of the single-channel and multi-channel beam steering results can be seen in Figure 8.

## 4. Experimental Setup

We present five experimental test cases to study the performance of the radar heart sound measurement, the stethoscope, and the accelerometer methods. Under all test case conditions, we had the subject sit still in a chair and breath normally. We secured the radar transceiver on a tripod in line with the front of the subject’s chest, approximately 0.5 m away. We used an elastic chest band to secure the stethoscope and accelerometer to the chest. All devices were connected to a laptop for data capture and signal processing. The measurements from all three methods were simultaneously collected and synchronized for direct comparison. We collected 60 s of data for each case. All measurements were conducted on one healthy, young, adult male.

### 4.1. Reference Case

To serve as a reference, we placed the stethoscope and accelerometer near the heart. The stethoscope was placed under the pectoral on the outer left side of the chest. The accelerometer was placed on the inner lower portion of the left pectoral. The chest band applied light pressure on the devices. This setup is depicted in Figure 9a.

### 4.2. Loose Fit Case

We tested the situation where the stethoscope and accelerometer are not securely in contact with the chest. We placed the stethoscope and accelerometer in the same location on the chest as the reference case. The chest band was worn loosely around the chest, only enough pressure was applied so that the devices did not fall off the chest. This setup is depicted in Figure 9b.

### 4.3. Tight Fit Case

We tested the case where considerably stronger pressure is applied to the stethoscope and accelerometer. We secured the chest band as tightly as possible while ensuring that the subject could still breath comfortably. The stethoscope and accelerometer were placed in the same location as the reference case. This setup is depicted in Figure 9c.

### 4.4. Thick Clothing Case

We tested the case where the subject is wearing thick clothing. We had the subject wear two additional sweatshirts in addition to the t-shirt they wore in the reference experiment. The stethoscope and accelerometer were otherwise configured exactly as they were in the reference case. This setup is depicted in Figure 9d.

### 4.5. Opposite Side Case

We tested the case where the device is secured on the side of the chest opposite the heart. The accelerometer was placed on the inner lower portion of the right pectoral. The stethoscope was placed under the pectoral on the outer right side of the chest. The chest band secured the devices in place with light pressure. This setup is depicted in Figure 9e.

### 4.6. Background Sound Case

We tested the case where there is significant background sound during data collection. We played loud music from a cell phone speaker at 65 dB, as measurement by a sound pressure sensor. As a reference, the sound pressure value is 40 dB for a quiet office room. The music was played approximately 0.5 m from both the subject’s chest and the radar device. The devices were otherwise configured as they were in the reference case. This setup is depicted in Figure 9f.

## 5. Data Analysis

### 5.1. Time–Frequency Analysis

We used the Short-time Fourier transform (STFT) to examine the spectral features the radar, stethoscope, and accelerometer. The STFT divides a longer time signal into shorter segments of equal length using a 70-point hamming windowing function with 64 overlapped samples to segment the signal [11]. The Fourier transform is then computed on each segment. We present the STFT as a spectrogram.

### 5.2. Signal-to-Noise Ratio (SNR) Calculation

We calculated the SNR of the radar, stethoscope, and accelerometer to determine their performance under each experimental case. We manually segmented the first heart sound (S1), the second heart sound (S2), and the noise for each signal. An example of the segmentation can be seen in Figure 10. We then took the power of each segment and averaged it over 40 cardiac cycles to calculate the ratio.

## 6. Results

We herein present the results of our studies. We analyzed the reference case signal’s time and frequency features for each measurement technique. We then calculated the signal-to-noise ratio for all three measurement techniques under each test case, and analyzed the results. The SNR results are depicted in Figure 11; numerical results can be found in Table 2. Further commentary will be made in the following subsections.

### 6.1. Reference Case

We analyzed the time domain of the reference case signals to examine the SNR of the three measurement methods. This can be seen in Figure 12. The SNRs of the first and second heart sound for the radar measurement are 10.9 dB and 10.7 dB, respectively. These are higher than the accelerometer signal (9.3 dB and 4.8 dB), but lower than the stethoscope signal (12.2 dB and 14.5 dB).

We here interpret the spectrogram of the reference case signals. The spectrogram can be found in Figure 13. The radar signal clearly separates the low-frequency components of the first and second heart sound from the noise floor. S1 and S2 are approximately 10 dB stronger than the noise floor. The radar fails to capture the higher-frequency components, both the signal and noise are contained below 75 Hz. The stethoscope captures both the high- and low-frequency components of the heart sounds. The signal is approximately 10–15 dB stronger than the noise floor. The accelerometer is only able to separate the low-frequency components of the heart sounds. Below 50 Hz, the heart sounds are 5–10 dB stronger than the noise. Above 50 Hz, the signal is no longer separable from the noise.

Under reference conditions, the stethoscope was able to capture the widest range of heart sound frequency components and separate the heart sound signals from the noise the most clearly. The radar and the accelerometer only captured the frequency components of signals below 75 Hz and 50 Hz, respectively. The radar measurements separated the heart sound signals from the noise better than the accelerometer.

### 6.2. Loose Fit Case

We here present the SNR results of the loose fit test case. The radar signal performed comparably, but slightly better (approximately 1 dB), than the radar reference case. The radar SNRs are 13.2 dB and 11.4 dB for S1 and S2, respectively. The stethoscope preformed considerably worse (approximately 10 dB) under the loose conditions compared to the stethoscope reference conditions. The stethoscope SNRs are 1.9 dB and 0.4 dB for S1 and S2, respectively. The accelerometer performed similar to its reference case. The accelerometer S1 SNR is 11 dB (1.7 dB higher than reference), while the S2 SNR is 4.8 dB (equal to reference).

The stethoscope performance degradation was expected because without secure contact, the stethoscope resonator cannot accurately capture the acoustic waves. We expected the accelerometer to perform markedly worse under the loose fit conditions. Our interpretation of these results is either the accelerometer was held more securely than we intended it to be, or our reference accelerometer fit was tighter than optimal. These results can be seen in Figure 14. We discuss the fluctuation in radar performance between test cases in Section 7.

#### Tight Fit Case

We here discuss the tight fit condition results. The radar S1 SNR is notably larger than the reference case at 16.1 dB (5.2 dB higher than reference). The radar S2 SNR is 11.6 dB (0.7 dB higher than reference). The stethoscope S1 and S2 SNRs are 7.3 dB and 5.9 dB, respectively. This is a significant reduction compared to the reference case, 4.9 dB lower for S1 and 8.6 dB for S2. The accelerometer also performed much worse than the reference. The S1 SNR is 4.7 dB lower than its reference, while the S2 SNR is 3.2 dB lower than the reference.

These results are in line with our expectations, as tight pressure from the chest band dampens skin vibrations under the stethoscope and accelerometer, affecting the signal quality. This result can be seen in Figure 14.

### 6.3. Thick Clothing Case

We now discuss the thick clothing test case. The radar S1 and S2 SNRs are 12.6 dB and 10.7 dB, respectively. This is comparable to the reference case (1.7 dB higher for S1 and equal for S2). The stethoscope performance is comparable to its reference case; the S1 SNR is 0.6 dB higher, while the S2 is 4.7 dB lower. The accelerometer signal was impacted by thick clothing; thus, the S1 SNR is 4.5 dB lower, while S2 is 1.3 dB lower, than the reference signal.

These results are believable and lead us to conclude that the accelerometer struggles to capture skin vibrations through thick clothing, while the stethoscope is able to overcome the barrier without much impact on signal quality. The radar is also able to penetrate thick clothing without reduction in SNR. This result is apparent in Figure 15. The noise floor is visually higher for the accelerometer, while the noise floor is comparable for the radar and stethoscope.

#### 6.3.1. Opposite Side Case

We analyzed the opposite side test case, and found that the radar performed better than it did in the reference case (4.6 dB higher for S1 and 1.1 higher for S2). The SNRs of the first and second radar heart sounds are 15.5 dB and 12.8 dB, respectively. The performance of the stethoscope is considerably worse than reference (3.8 dB lower for S1 and 6.9 dB for S2). The accelerometer performance is also markedly worse. The accelerometer S1 and S2 SNRs are 1.7 dB and 0.6 dB, respectively, making the heart sound signal almost indistinguishable from the noise.

The stethoscope and accelerometer results are expected, because the data were collected considerably further from the heart. We expected no reduction in radar SNR, because the radar’s field of view is much larger than the chest area. The improvement in radar SNR is noteworthy, and will be discussed further in Section 7.

#### 6.3.2. Background Sound Case

We now look at the results of the background sound case. The radar SNRs are 2 dB higher than the reference case for both S1 and S2. The accelerometer S1 SNR is 0.8 dB lower than reference, while the S2 SNR is 4.3 dB higher than the reference. We tested both the acoustic and digital stethoscope. The acoustic stethoscope performed notably worse than the reference case. The first and second radar heart sound SNRs are 8.6 dB and 7.6 dB, respectively (3.6 dB lower for S1 and 6.8 dB lower for S2). The digital stethoscope performed approximately 2 dB worse than the acoustic stethoscope for both S1 and S2.

As expected, we observed no degradation in the performance of the radar and the accelerometer under these conditions, as they should not capture background sound waves. The S2 SNR improvement of the accelerometer was not expected. We attribute this result to sensor placement and pressure discrepancy.

In addition to our radar, accelerometer, and acoustic stethoscope comparison, we want to highlight the difference between the acoustic and digital stethoscope performance in the background sound test. The performance degradation of the acoustic and digital stethoscope was expected, as the background sound should increase the noise power. This noise floor increase should be more prominent in the the digital case due to the signal amplification. This is in line with the results. This effect can be seen in Figure 16a. The noise increase is especially apparent at high frequencies (>200 Hz), as can be seen in Figure 16b.

## 7. Discussion

The proposed contactless stethoscope using radar outperformed the conventional accelerometer-based seismocardiogram heart sound measurement technique under the reference and all challenge conditions. The radar outperformed, or performed as well as, the stethoscope in terms of SNR under all challenge conditions. The radar measurement did not capture the higher-frequency components of the heart sound that the stethoscope was able to capture. We also observed variations up to 5 dB in the SNRs of the radar heart sound measurements between cases. We expected variation in SNR because the signal is influenced by the involuntary body motion and varying breathing patterns of the test subject. We hypothesize that an additional source of variation in the signal was caused by the weight and pressure of the reference stethoscope and accelerometer device attachment to the chest during the radar measurement, having a damping effect on the skin vibrations that the radar is capturing. The extent of the damping effect could vary depending on the stethoscope and accelerometer placement. A larger data set, collected over a diverse population, is needed before we can confidently identify the entire cause of variation in our data and better characterize the performance of the radar system.

## 8. Conclusions

We present a series of realistic challenge cases for our proposed contactless radar heart sound measurement, and compare it with conventional stethoscope and seismocardiogram measurement methods. We demonstrate our method’s ability to measure heart sound data in situations where conventional methods struggle. We use quantitative data to comment on the results of our studies. The contactless radar-based stethoscope outperformed conventional methods under several challenge scenarios, while offering the added comfort and convenience of contactless monitoring for patients and practitioners.

## Figures and Tables

**Figure 1 bioengineering-10-00169-f001:**
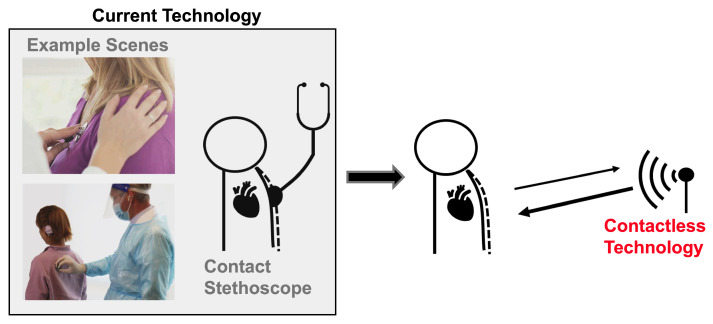
Current heart sound monitoring requires practitioners to attach a device onto a patient. Radar-based heart sound measurement provides an alternative measurement method that can be collected at a safe social distance, improving patient and a practitioner experience and permitting continuous monitoring.

**Figure 2 bioengineering-10-00169-f002:**
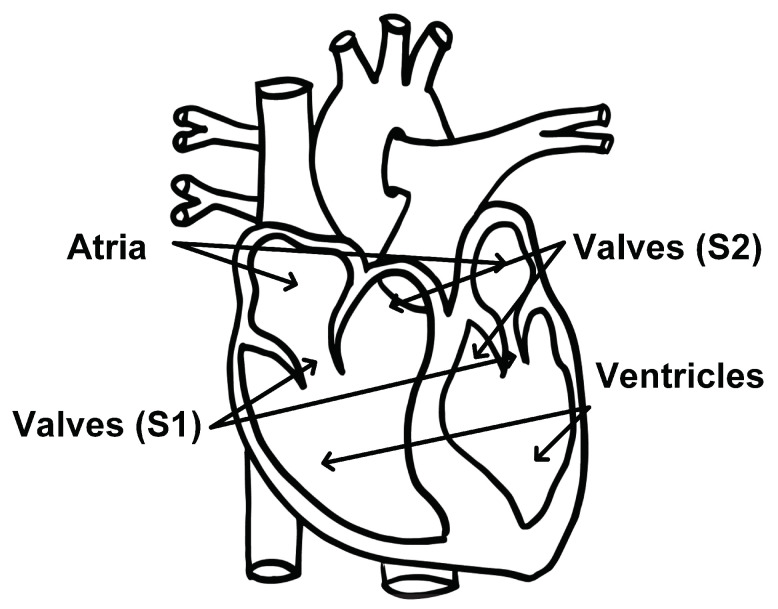
Diagram of heart components associated with the first and second heart sounds identified. Modified from [20].

**Figure 3 bioengineering-10-00169-f003:**
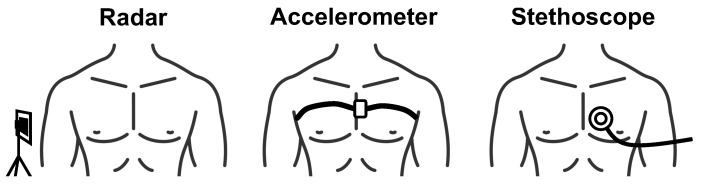
Depiction of contactless radar heart sound measurement compared with conventional heart sound measurement approaches.

**Figure 4 bioengineering-10-00169-f004:**
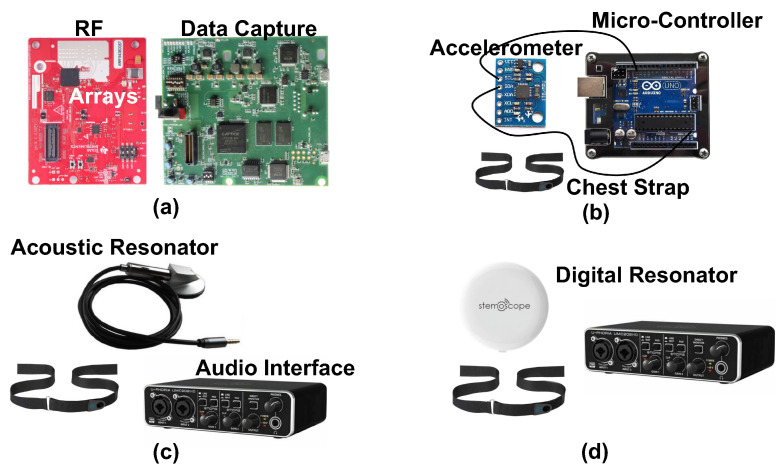
Photographs of the components of each of the four measurement systems used to capture heart sounds in our experiments. The radar system (**a**) consists of a radio frequency array and a data capture module. The accelerometer (**b**) is attached to the subject with a chest strap, and has a wired connection to a microcontroller for data capture. Both the acoustic (**c**) and digital (**d**) resonators attach to the chest with a chest strap. The acoustic resonator has a wired connection to the audio interface for data capture. The digital resonator wirelessly connects to a phone that then feeds the signal to the audio interface.

**Figure 5 bioengineering-10-00169-f005:**
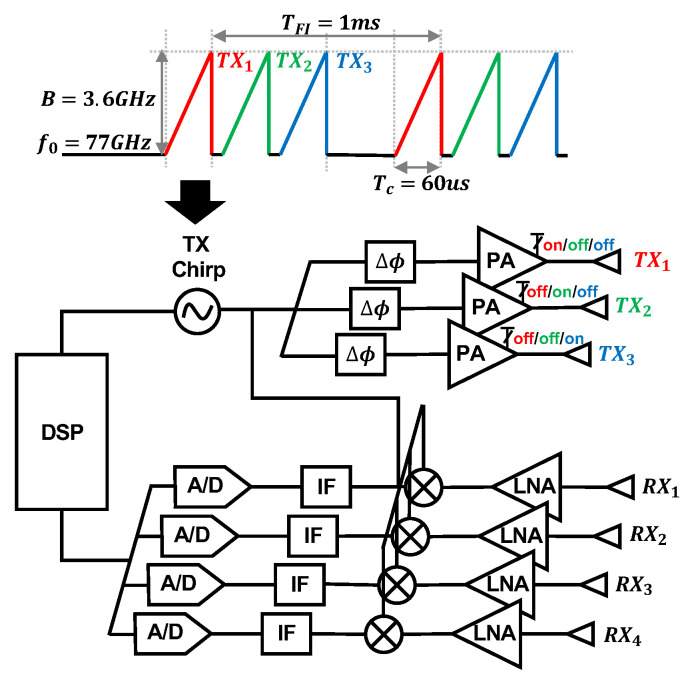
Block diagram of the Texas Instruments AWR6843 Radio Frequency (RF) array and the transmit waveform used in our study. The three transmit channels are colored. The signal is transmitted via TDM, meaning only one channel is transmitting at a given time, while the other channels are switched off. The four receive antennas are also labeled.

**Figure 6 bioengineering-10-00169-f006:**
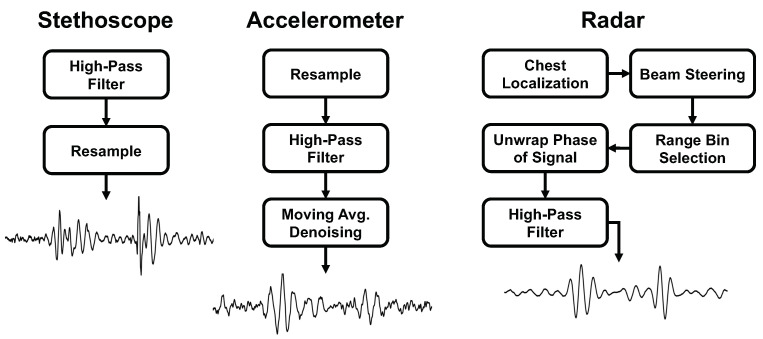
Comparison of the signal processing steps and final heart sound waveform for each of the measurement methods.

**Figure 7 bioengineering-10-00169-f007:**
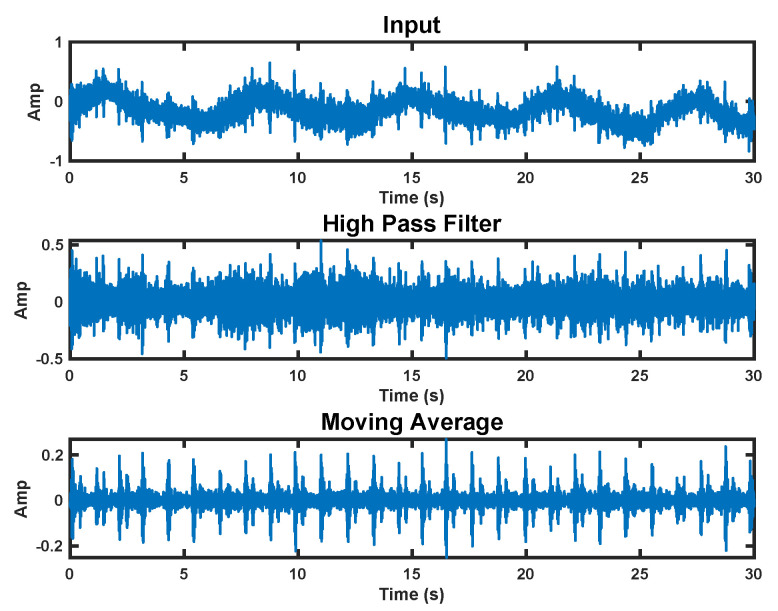
Accelerometer signal processing steps. The top plot shows the raw signal. The middle plot has a high−pass filter applied. The bottom plot is the final processed signal with a moving average applied.

**Figure 8 bioengineering-10-00169-f008:**
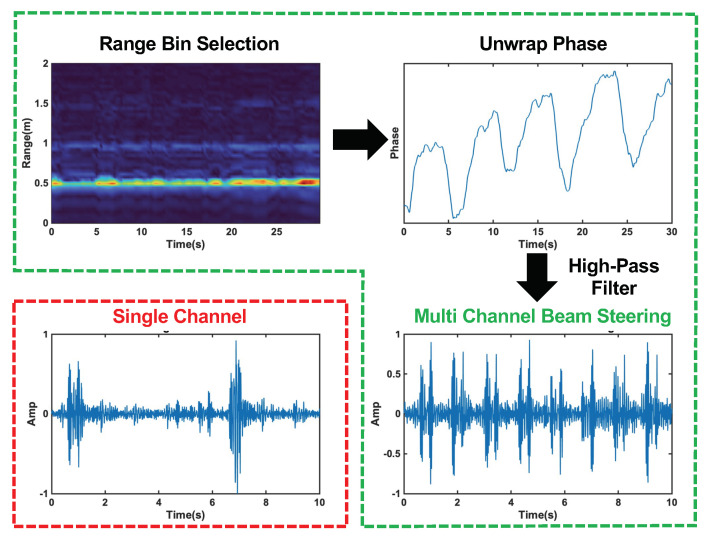
Radar signal processing steps with beam forming circled in green. The top left plot shows the range profile used for range bin selection. The top left plot displays the unwrapped phase of the signal at the selected range bin. The bottom right plots show the final heart sound signal after high−pass filtering for the multi−channel beam steering case, and the single−channel case is circled in red in the bottom left plot. The beam steering clearly captures each heart sound cycle, while the single−channel signal is distorted and misses many cycles.

**Figure 9 bioengineering-10-00169-f009:**
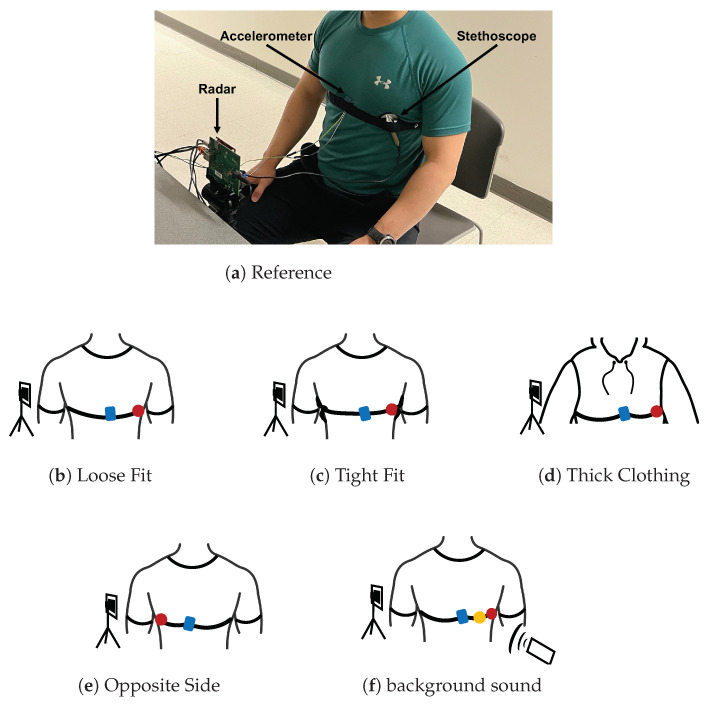
Photograph of the experimental setup under reference conditions with the three measurement devices labeled. A depiction of each test case configuration is provided below the photo. The accelerometer is presented in blue, the acoustic stethoscope in red, and the digital stethoscope in yellow.

**Figure 10 bioengineering-10-00169-f010:**
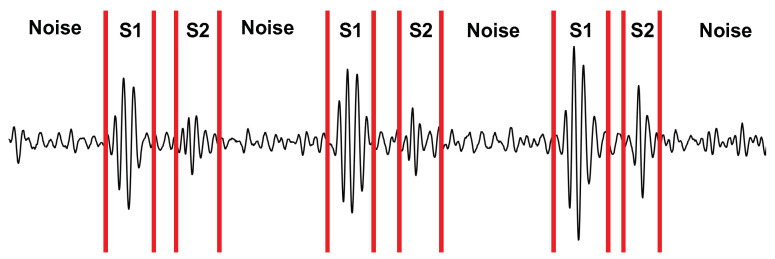
Depiction of the signal segmentation used for calculating the SNR.

**Figure 11 bioengineering-10-00169-f011:**
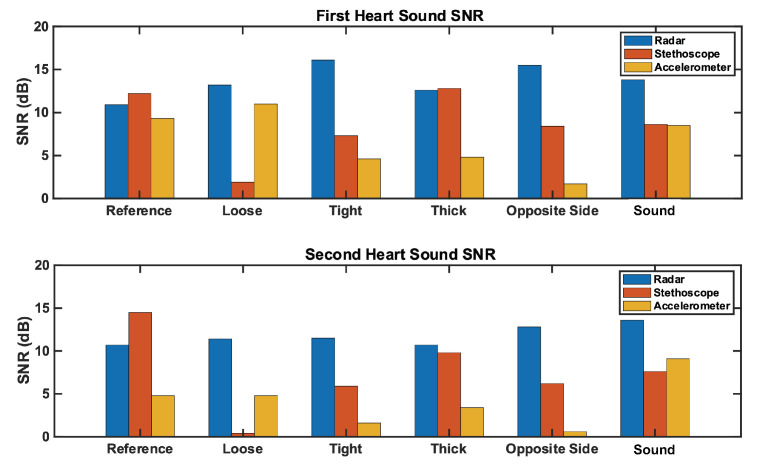
Comparison of the SNR for each measurement method across test cases. The (**top**) plot represents the first heart sound. The (**bottom**) plot represents the second heart sound. Radar SNR remains constantly high across test cases, while stethoscope and accelerometer SNRs vary significantly by test case.

**Figure 12 bioengineering-10-00169-f012:**
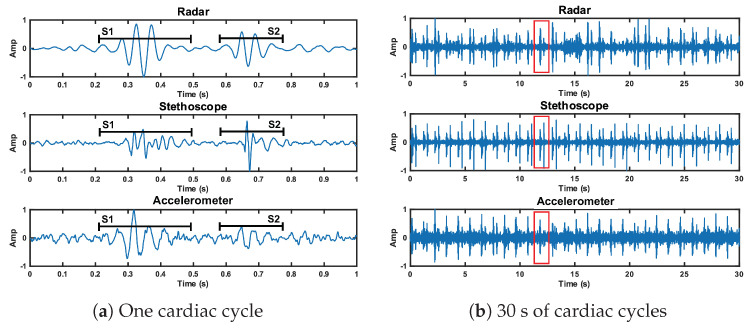
Time domain representation of the radar, stethoscope, and accelerometer measurement techniques. Plot (**a**) depicts a close up of one cardiac cycle, circled in red in plot (**b**).

**Figure 13 bioengineering-10-00169-f013:**
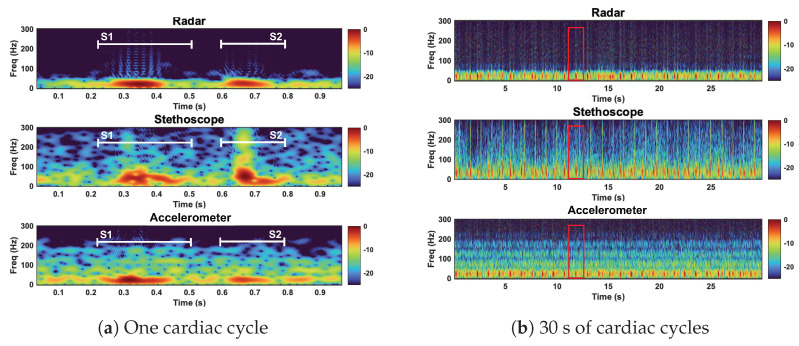
Spectrogram of the radar, stethoscope, and accelerometer measurement techniques. Plot (**a**) depicts a close up of one cardiac cycle, circled in red in plot (**b**). All three measurements capture the low−frequency components of the heart sounds. The stethoscope captures high−frequency heart sound features.

**Figure 14 bioengineering-10-00169-f014:**
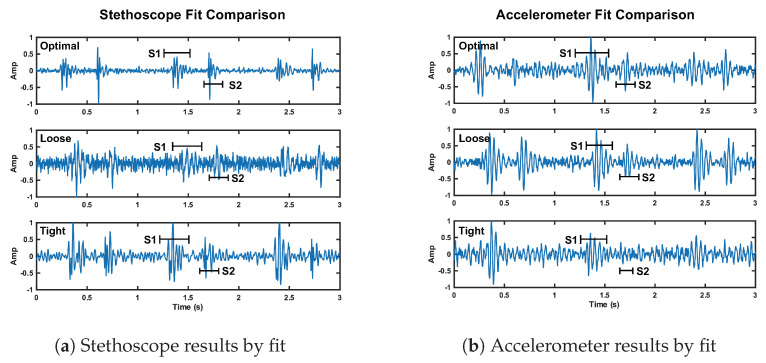
Depiction of the change in signal quality for the stethoscope (plot **a**) and accelerometer (plot **b**) over three cardiac cycles when the fit of the chest band is altered. The noise floor is visibly higher compared to reference fit for the stethoscope loose and tight fit, and the accelerometer tight fit.

**Figure 15 bioengineering-10-00169-f015:**
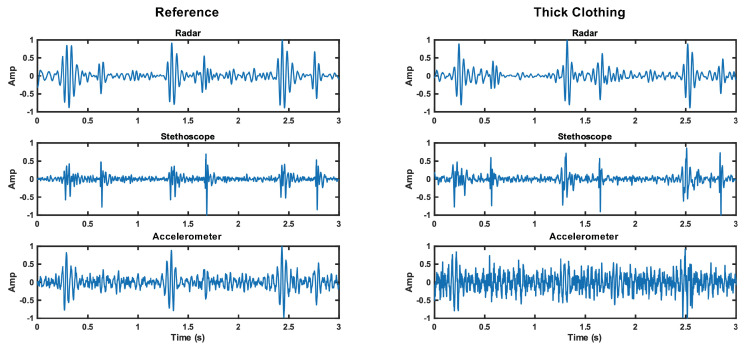
Comparison of the time domain waveforms for each measurement method under reference and thick clothing cases. The noise floor increase is apparent for the accelerometer.

**Figure 16 bioengineering-10-00169-f016:**
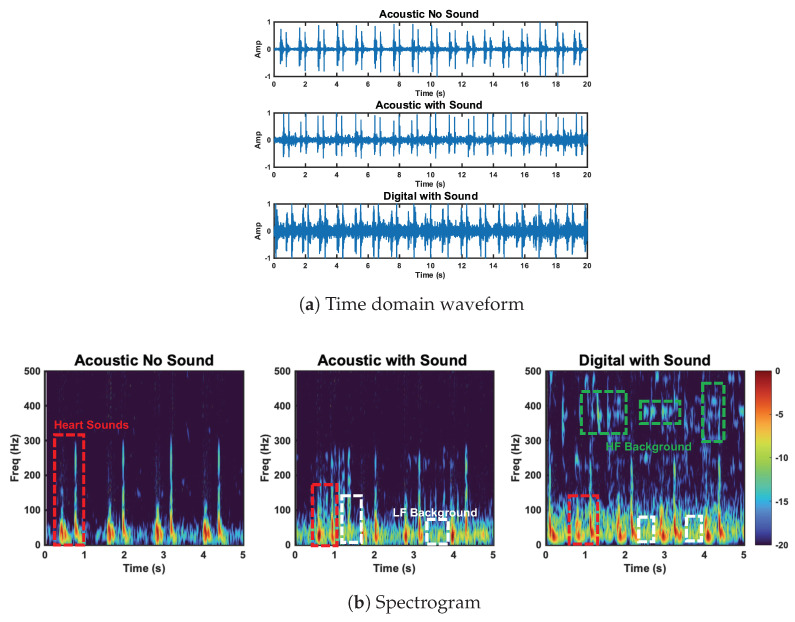
Comparison of the acoustic stethoscope without background sound, the acoustic stethoscope with background sound, and the digital stethoscope with background sound. Heart sound and high− and low−frequency background sounds are circled. Low−frequency background sounds are captured by the acoustic and digital stethoscope. HF background sounds are captured by the digital stethoscope.

**Table 1 bioengineering-10-00169-t001:** The key features of each measurement system.

Technology	Setup	Measurement	Robustness to Background Sound	Multiple Subjects Simultaneously	Clothes Penetration	Measurement Site	Costs
Radar	Radar Tripod	Remote Contactless	Excellent	Yes, via beam steering	Good	Entire chest	$850
Accelerometer	Accelerometer Microcontroller Chest Strap	Contact	Excellent	No	Poor	Small area near center chest subject to placement	$110
Acoustic Stethoscope	Chestpiece Audio Interface Chest Strap	Contact	Poor	No	Excellent	Strategic areas on chest or back	$170
Digital Stethoscope	Digital Chestpiece Audio Interface Chest Strap	Contact	Worst	No	Excellent	Strategic areas on chest or back	$200

**Table 2 bioengineering-10-00169-t002:** The SNRs for the first and second heart sounds for each measurement under each test case.

Case	Measurement	S1 SNR (dB)	S2 SNR (dB)
Optimal Fit	Radar	10.9	10.7
	Acoustic Stethoscope	12.2	14.5
	Accelerometer	9.3	4.8
Loose Fit	Radar	13.2	11.4
	Acoustic Stethoscope	1.9	0.4
	Accelerometer	11.0	4.8
Tight Fit	Radar	16.1	11.5
	Acoustic Stethoscope	7.3	5.9
	Accelerometer	4.6	1.6
Thick Clothes	Radar	12.6	10.7
	Acoustic Stethoscope	12.8	9.8
	Accelerometer	4.8	3.4
Opposite Side	Radar	15.5	12.8
	Acoustic Stethoscope	8.4	6.2
	Accelerometer	1.7	0.6
Background Sound	Radar	13.8	13.6
	Acoustic Stethoscope	8.6	7.6
	Digital Stethoscope	6.2	6.5
	Accelerometer	8.5	9.1

## Data Availability

Data available on request due to test subject privacy restrictions.

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
