# Peer review of "Contactless Stethoscope Enabled by Radar Technology"

_bioengineering, 2023, doi:10.3390/bioengineering10020169_

Round 1
Reviewer 1 Report
the abstract not present the advantage of the proposed system , a comparison table should be presented clearly the difference between three system for heart rate extraction and another comparison parameter should be added as cost, easy of measured and extraction
the paper should be reorganized to show the novelty of the proposed system and advantage
the organization of the paper should be clear and English way what you want from the comparison
Reviewer 2 Report
I enjoyed reading the manuscript Contactless Stethoscope Enabled by Radar Technology by Lenz et al. In it, the authors are presenting the design, and verification, of a stethoscope based on mm-wave FMCW radar, including a hardware design and signal processing required to extract the audio (?) signal of the heart valves.
While I enjoyed the paper, I have several issues with it.
- First, the hardware part is glanced over, and mostly covered in figure 4, which does not qualify for an explanation of circuitry - at least a diagram would be required for that, and part numbers for the bigger modules used.
- Second, given that the authors want to operate something that uses RF radiation practically in contact with the body, some sort of safety argument, and SAR calculations should be provided.
- The authors present data from a human study, but there is no mention whatsoever of IRB approval. This needs to be provided, or for ethical reasons all data would be disqualified.
- Continuing on the human study design, there is no information about the number of subjects, inclusion/exclusion criteria. How many subjects were the results based on? Were the subjects standing still during the data acquisition? I am also missing any sort of statistics.
- The authors claim that this is a contactless design, and part of their results seems to indeed point to an operation from some distance. However, most of their results come from application from either in contact with the body or attached over clothes, which hardly qualifies as contactless.
- After reading the paper, I am still wondering what exactly the need is that the authors are trying to address. I can see that the radar technology presented outperforms the other two candidates for the strapped-on design (but there, you need to convince me of its safety). Why is remote monitoring of heart sounds needed? And how does that part perform?
I think my main problem with this paper, as it stands, is that the presentation of the human study seems to be disregarding ethical standards for human subject research (IRB approval, safety considerations). I hope this part is just a lapse in writing that can be addressed in a revised version of this manuscript.
Reviewer 3 Report
The topic of this paper is quite interesting, the paper is well written and organized but some issues should be clarified. The main reviewer concerns are summarized as follows
· It is not clear what is new regarding [25] and other previous publications of the same authors. Could you please explain it better?
· The reviewer suggest to add a paragraph in the end of introduction with the organization of the rest of paper.
· Have you just used a commercial RADAR? Or have you also implemented some signal processing algorithm?
· How many antennas have you used and what beamforming algorithm have you considered?
Round 2
Reviewer 3 Report
The authors have satisfactory addressed the main reviewer concerns. The paper is now in conditions to be accepted.
Author Response
We thank the reviewer for their comments